# Microencapsulation of *Camellia oleifera* Seed Oil Emulsion By-Products: Structural Characterization and Lipidomics Analysis

**DOI:** 10.3390/foods14193314

**Published:** 2025-09-24

**Authors:** Xue Wu, Yunhe Chang, Mingfa Song, Juncai Hou, Hongxia Feng

**Affiliations:** 1School of Food Science and Engineering, Guiyang University, Guiyang 550005, China; 18224722993@163.com (X.W.); changyunhe1102@sina.com (Y.C.); 18302685242@163.com (M.S.); houjuncai88@126.com (J.H.); 2Engineering Technology Research Center for Processing and Comprehensive Utilization of Idesia Polycarpa of National Forestry and Grassland Administration, Guiyang 550005, China

**Keywords:** *Camellia oleifera* seed oil, aqueous enzymatic extraction, emulsion-phase microcapsules, characterization, lipidomics

## Abstract

To unlock the potential value of the emulsified by-product from the aqueous enzymatic extraction (AEE) of Camellia oleifera seed oil, this study introduced an innovative approach for its food industrial application. We applied spray-drying microencapsulation technology to convert the emulsion-phase (EP) by-product into value-added microcapsules (EPM). The properties of EPM were systematically compared with those of microcapsules derived from the oil phase (OPM). The encapsulation efficiencies of EPM and OPM were 83.94% and 86.53%, respectively. Scanning electron microscopy revealed the formation of irregular spheroids with smooth surfaces and intact structures, with EPM exhibiting superior particle uniformity (D_50_ = 1.11 μm) compared to OPM (D_50_ = 2.30 μm). Fourier-transform infrared spectroscopy confirmed the successful encapsulation of EP. Differential scanning calorimetry indicated good thermal stability of the microcapsules, and the oxidative stability of EPM (24.75 h) was 9.2 times higher than that of the unencapsulated EP and 13.15 h longer than that of OPM. Lipidomic analysis using LC-MS/MS identified 477 lipid species across five subclasses—glycerolipids, glycerophospholipids, fatty acids, prenol lipids, and sphingolipids—revealing distinct lipid profiles between EPM and OPM. This microencapsulation strategy offers a sustainable approach to valorize aqueous enzymatic extraction by-products, with promising applications in functional foods and nutraceuticals, thereby enhancing the economic and environmental sustainability of *Camellia oleifera* seed oil processing.

## 1. Introduction

*Camellia oleifera* Abel., a member of the Theaceae family, is recognized as one of the world’s four major woody oil crops, alongside olive, oil palm, and coconut [1]. It is widely cultivated in countries such as China, the Philippines, Thailand, Japan, and South Korea [2]. The seeds of *Camellia oleifera* have a high oil content (24.3–59.7%), surpassing that of many conventional oil seeds [3]. Camellia seed oil is particularly rich in unsaturated fatty acids (UFAs), which account for more than 90% of its total fatty acid composition. Notably, it contains the highest concentration of oleic acid among commonly consumed edible oils [4]. In addition to its favorable fatty acid profile, camellia seed oil is a source of various bioactive compounds, including polyphenols, sterols, tocopherols, and squalene [5]. These constituents exhibit a range of health-promoting biological activities, such as antioxidant effects, liver protection, anti-inflammatory properties, anticancer potential, and relief of gastrointestinal discomfort [1].

Currently, the industrial-scale extraction of camellia seed oil primarily relies on mechanical pressing extraction (MPE) and solvent extraction (SE), which are widely adopted due to their low capital investment and operational simplicity [6]. However, these conventional methods present notable drawbacks, including reduced oil quality caused by the presence of hazardous solvent residues and the need for extensive refining [7]. To address these limitations, alternative extraction techniques such as supercritical fluid extraction (SCFE) and aqueous extraction have been developed. Despite their technical advantages, these methods face significant commercialization barriers, including operational complexity and high capital costs [8]. Consequently, there remains a need to further optimize industrial processing techniques for camellia seeds to meet increasing market demand. Aqueous enzymatic extraction (AEE) has emerged as a promising alternative, offering several advantages such as mild processing conditions, high oil yield, and technological feasibility. Studies have shown that oils extracted via enzymatic methods contain higher levels of beneficial nutrients—such as tocopherols and phenolics—and exhibit superior oxidative stability compared to hexane-extracted oils [5]. Gai et al. [9] reported that oil obtained through AEE (AEEO) displayed physicochemical properties comparable to those of Soxhlet-extracted oil (SEO), while demonstrating enhanced oxidative stability, as evidenced by lower levels of conjugated dienes, peroxide value, and anisidine value. Additionally, AEEO was found to contain higher concentrations of valuable polyunsaturated fatty acids and tocopherols, and showed improved DPPH radical scavenging activity [4].

The aqueous enzymatic method has been applied to the industrial production of camellia seed oil. However, during the extraction process, in addition to the recovery of the oil phase, several by-products are generated, including the emulsion phase, aqueous phase, and solid residue [10]. These by-products are rich in bioactive compounds such as high-quality saponins, polyphenols, and sterols. Current research has largely focused on demulsification and recovery of the aqueous phase [11], while the emulsion phase itself remains underexplored for value-added applications.

Several studies have demonstrated the potential of utilizing by-products from aqueous extraction processes. For instance, soy protein hydrolysate (SPH) from enzyme-assisted soybean oil extraction was successfully microencapsulated, improving its antioxidant activity and solubility [12]. Similarly, rapeseed protein isolate (RPI) has been used as a wall material for peptide encapsulation [13], and liquid by-products from enzyme-assisted aqueous extraction processing (EAEP) have been spray-dried to produce polypeptide-rich powders. Moreover, enzymatically modified biopolymers, such as protease-hydrolyzed maltodextrin combined with rice protein isolate, have significantly enhanced the oxidative stability of microencapsulated linseed oil [14].

While considerable efforts have been devoted to demulsification and compound separation from AEE by-products, limited attention has been paid to the direct utilization of the emulsion phase (EP)—a resource-rich mixture of oil, proteins, phospholipids, and saponins—as a functional material. We hypothesize that the EP can serve as a dual-functional core material for microencapsulation, leveraging its inherent emulsifying and bioactive components to enhance both oxidative stability and nutritional functionality.

This study introduces an innovative approach to valorizing the emulsion phase by-product from AEE of camellia seed oil through spray-drying microencapsulation. The specific objectives are: (1) utilize the emulsion-phase by-product as a dual-functional core material and apply spray-drying microencapsulation technology; (2) to characterize the physicochemical, structural, and thermal properties of EPM and OPM; (3) to evaluate the oxidative stability and lipidomic profiles of the microencapsulated products; (4) to assess the potential of EPM as a functional ingredient for lipid-based formulations in foods and nutraceuticals. The spray-drying process preserves native phytochemicals such as phenolics and saponins, while simultaneously encapsulating a substantial oil payload. This conversion of liquid lipids into a stable, powder-based matrix significantly enhances oxidative stability. This strategy not only contributes to sustainable utilization of camellia seed processing by-products but also offers a new pathway for designing high-value, stable lipid ingredients with retained bioactivity.

## 2. Materials and Methods

### 2.1. Raw Materials and Reagents

Camellia seeds (*C. oleifera* Abel.) were obtained from a local forest farm in Ceheng City, China. Alkaline protease (EC 3.4.21.62) was procured from Aladdin Reagent Co., Ltd. (Shanghai, China). Food-grade soybean protein isolate (SPI, ≥90% protein) and maltodextrin (DE 10-12) were obtained from Henan Qianzhi Co., Ltd., Henan, China. Sucrose fatty acid ester and glycerol monostearate (all food grade) were purchased from Shanghai Yuanye Bio-Technology Co., Ltd., China (Manufacturer: Shanghai Yuanye Bio-Technology Co., Ltd.; City: Shanghai; Country: China). Potassium bromide (KBr), which is analytically pure, was acquired from MACKLIN Co., Ltd., China (Manufacturer: MACKLIN Co., Ltd.; City: Shanghai; Country: China). The reagents such as ethanol, ether and petroleum ether are all of analytical purity and were purchased from Huadong Medicine Co., Ltd., Hangzhou, China. The water used in the experiments was deionized water.

### 2.2. Preparation of Microcapsule

#### 2.2.1. Preparation of the Oil and Emulsion Phases of *Camellia oleifera* Seed

The oil and emulsion phases were prepared following the method of Peng Li et al. [15], with slight modifications. The detailed procedure for the aqueous enzymatic extraction of *Camellia oleifera* seed oil was as follows: (1) Fresh *Camellia oleifera* seeds were sun-dried and stored at room temperature under dry conditions. After manual shelling, the seeds were ground using a grinder (BF-08, BenChen, Hebei, China) and passed through a 0.85 mm sieve to obtain a homogeneous powder. (2) A measured amount of the seed powder was mixed with water at a ratio of 1:6 (*w*/*v*). (3) The mixture was incubated in a temperature-controlled magnetic stirring system. Before enzymatic hydrolysis (Alcalase 2.4 L), the pH was adjusted to the desired value using 0.5 M NaOH, followed by the addition of 1% (*w*/*w*) enzyme solution. (4) Hydrolysis was carried out under continuous stirring at 50 °C for 2 h. The resulting slurry was centrifuged (KH20R, Shanghai, China) at 8000 rpm for 30 min, resulting in four distinct layers (from top to bottom): oil phase, emulsion phase, aqueous phase, and solid residue. The oil and emulsion phases were carefully collected for subsequent microencapsulation.

#### 2.2.2. Microencapsulation of Oil and Emulsion Phases

The oil and emulsion phases obtained from the previous extraction steps were used as core materials, following a modified method based on Hu et al. [16]. First, a composite emulsifier system—comprising sucrose fatty acid ester and glycerol monostearate in a 1:4 mass ratio—was dissolved in ultrapure water at 65 °C to prepare a 1.4% (*w*/*w*) emulsifier solution. Soy protein isolate (SPI) and maltodextrin (MD), mixed at a 1:1 (g/g) ratio as composite wall materials, were then added to the emulsifier solution and stirred continuously for 30 min to ensure complete dissolution. The core material (oil phase or emulsion phase) was subsequently added at a core-to-wall ratio of 3:5 (g/g), followed by an additional 30 min of homogenization to achieve a final solid content of 15%. The mixture was then subjected to high-shear emulsification (AD300L-H, Shanghai, China) at 10,000 rpm and three cycles of high-pressure homogenization at 30–40 MPa to stabilize the emulsion. For spray drying (YC-510, Shanghai, China), the operational parameters were as follows: inlet temperature of 180 °C, outlet temperature maintained between 70 and 80 °C, pump speed at 50%, fan airflow at 100%, and nozzle airflow adjusted to 30–40 m^3^/h. The emulsion was passed through the nozzle three times. The resulting microcapsules were classified into two groups based on the core material used: oil-phase microcapsules (OPM) and emulsion-phase microcapsules (EPM).

### 2.3. Encapsulation Efficiency (EE)

#### 2.3.1. Determination of Surface Oil Content

The surface oil content of the microcapsules was determined according to the method described by Hu et al. [16]. The procedure involved three successive extractions using n-hexane at a material-to-solvent ratio of 1:30 (*m*/*v*), with each extraction lasting 2 min. After extraction, the solvent was removed via rotary evaporation, and the remaining residue was dried to a constant weight. The surface oil content was then calculated based on the weight of the dried residue.(1)$$surface\;oil\;content=\frac{M_{b}-M_{a}}{M}\times 100\%$$
where M: Weigh the mass of the sample, g; M_a_: The quality of the flask, g; M_b_: The mass difference between the dried sample and the flask, g.

#### 2.3.2. Determination of Total Oil Content

The total oil content and encapsulation efficiency (EE) of the microcapsules were determined using a modified method based on Cui et al. [17]. Briefly, a sample mass (m) of 4 g was accurately weighed into a beaker. Then, 20 mL of deionized water (preheated to 60 °C) and 2 mL of 25% ammonia solution were added to dissolve the sample under continuous stirring at 50 °C for 30 min. Subsequently, 10 mL of ethanol and 25 mL of diethyl ether were added, and the mixture was gently vortexed for 5 min. Next, 25 mL of petroleum ether was introduced, and the solution was gently agitated for another 5 min, followed by a 5 min rest to allow phase separation. The upper organic phase was carefully collected, and the lower aqueous phase was re-extracted twice using a ternary solvent mixture of ethanol, diethyl ether, and petroleum ether (1:1:1, *v*/*v*/*v*). The combined organic extracts were evaporated to dryness under reduced pressure, dried to constant weight in a desiccator, and precisely weighed (M). The total oil content of the microcapsules was calculated using Equation (1), and the encapsulation efficiency (EE) was determined as the ratio of encapsulated oil to total oil content according to Equation (2).(2)$$Total\;oil\;content\;(\%)=\frac{M_{1}-M_{2}}{m}\times 100\%$$(3)$$EE\;(\%)=\frac{Total\;oil\;content-Surface\;oil\;content}{Total\;oil\;content}\times 100\%$$
where m: Sample mass, g; M_1_: Mass of the rotary evaporation bottle and the total oil, g; M_2_: Mass of the rotary evaporation bottle, g.

### 2.4. Morphology Characterization of Microcapsule

#### 2.4.1. Moisture Content

The moisture content of the encapsulated powder was measured following the AOAC 925.10 gravimetric method. Approximately 2 g of sample (accurately weighed to ±0.0001 g) was placed in a pre-dried and weighed aluminum dish and dried in a hot-air oven (Model DHG-9140A, Yiheng Technical Co., Ltd., Shanghai, China) at 105 ± 2 °C until a constant weight was achieved (typically after 4–5 h). The moisture content was calculated using the following equation:(4)$$Moisture\;content=\;\frac{M_{1}-M_{2}}{M}\times 100\%$$
where M_1_ is the initial weight of the sample (g), and M_2_ is the weight after drying (g). All measurements were performed in triplicate, and the results are expressed as mean ± standard deviation.

#### 2.4.2. Optical Microscopy

The microcapsules were prepared as smears, observed under an optical microscope (BX53, OLYMPUS, Tokyo, Japan), and photographed [18].

#### 2.4.3. Morphology Observation (SEM)

Sample preparation for scanning electron microscopy (SEM) followed a standardized protocol with modifications [19]. Briefly, double-sided conductive adhesive tape was affixed to the sample stage, and microcapsule powder was evenly spread onto the adhesive surface. Excess particles were gently removed using a rubber air bulb to isolate individual particles. The samples were then sputter-coated with gold for 90 s under an argon atmosphere to enhance surface conductivity. Morphological evaluation was performed using a field-emission SEM (SU8600, HITACHI, Tokyo, Japan) at an accelerating voltage of 1 kV and a working distance of 10.6 mm. Images were captured at magnifications of 1000× and 10,000× to assess particle integrity at the macroscopic level and surface topography at the nanoscale.

#### 2.4.4. Dispersibility

The dispersibility of the microcapsule samples in cold water was determined using a modified method based on the procedure described by Peng et al. [20]. Briefly, 0.5 g of microcapsules was mixed with 12 mL of distilled water and stirred vigorously for 2 min. The mixture was then centrifuged at 1500 rpm for 15 min. After centrifugation, 5 mL of the supernatant was carefully collected, transferred to a Petri dish, and dried at 105 °C for 6 h. The weight of the dried solid residue obtained from the 5 mL supernatant aliquot was recorded as M (g). The solubility of the microcapsules was calculated according to Equation (5).(5)$$Dispersibility\;(\%)=\frac{grams\;of\;powder\;in\;the\;petri\;dish\times 3}{total\;grams\;of\;powder}\times 100\%$$

#### 2.4.5. Powder Flow Properties Analysis

##### The Carr Index and the Hausner Ratio

The flowability of the microcapsule powders was comprehensively evaluated by measuring the angle of repose, the Carr index, and the Hausner ratio. The Carr index and Hausner ratio were determined according to the standard methods described in the United States Pharmacopeia<1174> [21]. Briefly, the bulk density (ρ_bulk) and tapped density (ρ_tapped) were measured using a graduated cylinder and a tapped density tester (Model XYZ). The Carr index (CI, %) and Hausner ratio (HR) were calculated using the following equations:(6)$$CI\left(\%\right)=\;\frac{\rho_{tapped}-\rho_{bulk}}{\rho_{tapped}}\times 100$$(7)$$HR=\frac{\rho_{tapped}}{\rho_{bulk}}$$

##### Repose Angle

According to Cano-Chauca et al. [22], the angle of repose (θ) is used to assess the fluidity of powders, an important parameter of product quality. Briefly, 1 g of microcapsule powder was poured vertically through a funnel onto a clean, horizontal plastic plate, forming a natural conical pile. The angle of repose (θ) was calculated by measuring the pile height (h) and base radius (r) using Equation (8). A smaller angle of repose indicates better powder flowability, as the angle is inversely related to flow characteristics.(8)$$Repose\;Angle\;(\theta )=arctan\frac{h}{r}$$
where h: the pile height; r: base radius.

#### 2.4.6. Particle Size Distribution and Zeta Potential

The particle size and zeta potential of the microcapsule samples were characterized using a laser particle size analyzer (Dylisizer NS-90Z, OMEC, Zhuhai, China). Particle size distribution was quantified by calculating both the weighted average diameter and the size range, while surface charge characteristics were assessed via zeta potential measurements at different pH levels. Sample preparation followed an optimized protocol [23], where aliquots of a homogeneous microcapsule suspension were diluted with deionized water at a ratio of 1:100 (*v*/*v*). For particle size analysis, 50 μL aliquots were introduced into a quartz cuvette using precision micropipettes, and dispersion was ensured by vortex mixing three times for 10 s each. For zeta potential analysis, the pH of the samples was adjusted between 4 and 9 at room temperature using 0.1 M NaOH or HCl solutions. At each pH interval (ΔpH = 1), 1 mL samples were collected and measured at 25 ± 0.5 °C.

### 2.5. Fourier Transform Infrared Spectroscopy Analysis

The Fourier transform infrared (FTIR) spectra of the samples were recorded following the method reported by Peng et al. [20]. Briefly, the sample was mixed with potassium bromide (KBr) powder, pressed into tablets, and then scanned using an FTIR spectrometer (EQUINOX 55, BRUKER, Billerica, MA, USA) at a resolution of 4 cm^−1^ over a wavelength range of 400–4000 cm^−1^.

### 2.6. Differential Scanning Calorimetry (DSC)

DSC was used to evaluate the thermal properties of the microcapsule samples following the method described by Székely-Szentmiklósi et al. [19]. Approximately 4.0 mg (±0.01 mg) of each sample was accurately weighed into 50 µL sealed alumina crucibles. The crucibles were heated from 25 °C to 250 °C at a constant rate of 10 °C/min under a nitrogen atmosphere. An empty alumina crucible served as the reference. Melting curves were recorded and analyzed.

### 2.7. Thermogravimetric Analysis (TGA)

The thermal stability and decomposition profile of the microcapsules were investigated by thermogravimetric analysis (TGA) using a simultaneous thermal analyzer (HITACHI STA200, Hitachi, Tokyo, Japan). Approximately 5–10 mg of each sample (OPM and EPM) was accurately weighed into a standard alumina crucible. An empty alumina crucible was used as a reference. The temperature program was set from 25 °C to 50 °C at a constant heating rate of 10 °C/min [24].

### 2.8. Oxidation Stability of Microcapsules

The oxidation stability index of the samples was measured according to the method reported by Timilsena et al. [25]. Using an oxidation stabilizer (JF14112B, Hunan, China), thermogravimetric and differential scanning curves of OP, EP, OPM, and EPM samples were obtained to assess their oxidation stability. After instrument stabilization, 3 g of each sample was weighed for analysis.

### 2.9. Lipid Composition

#### 2.9.1. Sample Preparation and Extraction

The lipid extraction was performed according to the methyl-tert-butyl ether (MTBE)-based method [26] with modifications optimized for microencapsulated samples. Briefly, Samples stored at −80 °C were thawed on ice. Then, 20 mg of sample was weighed and mixed with 1 mL of extraction solvent (MTBE: MeOH = 3:1, *v*/*v*) containing an internal standard mixture. The mixture was vortexed for 15 min, followed by the addition of 300 μL ultrapure water. After vortexing for 1 min, the sample was centrifuged at 12,000 rpm for 10 min. A 100 μL aliquot of the upper organic phase was collected and evaporated using a vacuum concentrator. The dry residue was reconstituted in 200 μL of solvent (ACN: IPA = 1:1, *v*/*v*) prior to LC-MS/MS (QTRAP 6500+, SCIEX, Boston, MA, USA) analysis.

#### 2.9.2. Chromatography–Mass Spectrometry Acquisition Conditions

Lipidomics analysis was performed following the method of Zhihui et al. [27] with slight modifications. The data acquisition system comprised an ultra-high-performance liquid chromatography (UHPLC) system (ExionLC™ AD, SCIEX, Framingham, MA, USA) coupled with a tandem mass spectrometer (QTRAP^®^ 6500+, SCIEX, Framingham, MA, USA).

UPLC conditions: The chromatographic separation was performed using a Thermo Accucore™ C30 column (2.6 μm, 2.1 mm × 100 mm i.d.) maintained at 45 °C with a flow rate of 0.35 mL/min. The mobile phase consisted of two solvents: Phase A was acetonitrile/water (60:40, *v*/*v*) containing 0.1% formic acid and 10 mmol/L ammonium formate, while Phase B was acetonitrile/isopropanol (10:90, *v*/*v*) with the same additives. Gradient elution was carried out with the following proportions of Phase A to Phase B (*v*/*v*): 0 min, 80:20; 2 min, 70:30; 4 min, 40:60; 9 min, 15:85; 14 min, 10:90; 15.5 min, 5:95; 17.3 min, 5:95; 17.5 min, 80:20; and 20 min, 80:20.

MS conditions: Electrospray ionization (ESI) source temperature was set at 500 °C. The ion spray voltage was 5500 V in positive ion mode and −4500 V in negative ion mode. Ion source gases were set as follows: Gas 1 (GS1) at 45 psi, Gas 2 (GS2) at 55 psi, and Curtain Gas (CUR) at 35 psi. Each ion pair was scanned and detected in the triple quadrupole mass spectrometer using optimized Declustering Potential (DP) and Collision Energy (CE) settings.

#### 2.9.3. Lipid Content Determination

Quantification of lipids was performed using a targeted internal standard-based approach, a widely accepted method in lipidomics [28]. The mass spectral data were analyzed using Analyst software version 1.6.3. Lipid identification was performed by referencing the LIPID MAPS^®^ database and a local spectral library. The content of each lipid was calculated using the following formula:(9)$$X\left(\%\right)=\frac{0.001\times R\times c\times F\times V}{m}\times 100\%$$

X: lipid content in the sample (n mol/g);

R: the peak Area Ratio of the substance to the internal standard peak area;

F: correction coefficient of internal standard for different types of substances;

c: internal standard concentration (μmol/L);

V: sample extraction solution (μL);

m: sample size taken (g).

### 2.10. Statistical Analysis

All experiments in this study were conducted with a minimum of three independent biological replicates (*n* ≥ 3). Data are presented as mean ± standard deviation (SD). Statistical analyses were performed using IBM SPSS Statistics (Version 26.0, IBM Corp., Armonk, NY, USA). The significance of differences was determined by an independent samples Student’s *t*-test. A *p*-value of less than 0.05 (*p* < 0.05) was considered statistically significant. All graphs and figures were generated using OriginPro (Version 2018, OriginLab Corp., Northampton, MA, USA). Lipidomics data processing and multivariate statistical analysis were conducted using the following software and platforms: raw mass spectrometry data were processed using LipidSearch™ software (Version 4.2, Thermo Fisher Scientific, Waltham, MA, USA) for peak alignment, lipid identification, and quantification. Subsequent multivariate analysis, including Principal Component Analysis (PCA), was performed using SIMCA^®^ software (Version 16.0.2, Sartorius Stedim Data Analytics AB, Umeå, Sweden).

## 3. Results and Discussion

### 3.1. Physical Properties of OPM and EPM

This study compared two types of microcapsules—oil-phase microcapsules (OPM) and emulsion-phase microcapsules (EPM)—to evaluate their suitability for food applications, revealing key differences in their physical properties (Table 1). COM was a milky white powder, whereas EPM exhibited a light grayish-white color. Both were odorless, impurity-free, loose, and showed no caking. The darker color of EPM is normal and originates from the light brown color of the oxidized AEEE raw material. No other sensory defects were observed in EPM. The color parameters *L**, *a**, and *b** were used to characterize the appearance of the microcapsules. The significantly higher *L** value (brightness) of EPM compared to OPM (*p* < 0.05) indicates greater luminosity and light transmittance in the emulsion-phase microcapsules. This finding aligns with Zhao et al. [29], who reported that using soy protein isolate and maltodextrin as wall materials enhances microcapsule transparency. Compared to OPM, the *a** value (redness) of EPM increased significantly, while the *b** value (yellowness) remained largely unchanged. This may be attributed to trace amounts of protein degradation products or polysaccharide fragments formed during enzymatic hydrolysis in the emulsion phase, which impart a slightly reddish hue and increased translucency to EPM [30].

As shown in Table 1, the encapsulation efficiency of camellia seed oil microcapsules (OPM) and emulsion-phase microcapsules (EPM) was 86.53% and 83.94%, respectively, indicating nearly complete encapsulation of the core materials. The slightly lower encapsulation efficiency of EPM may be due to its core material (emulsion phase) containing only 61.83% total oil content. This is consistent with Yang J. Y. et al. [31], who reported approximately 67.83% crude fat in emulsified oils.

Both microcapsule types achieved encapsulation efficiencies above 80%, but EPM exhibited significantly higher water dispersibility (90.50%) compared to OPM (57.68%). This enhanced emulsification and solubility are likely due to the presence of surfactants (tea saponin) in the emulsified camellia seed layer [32], making EPM particularly suitable as a functional food ingredient in industrial applications.

The moisture contents of OPM and EPM were 1.29% and 1.88%, respectively, with low moisture levels favoring longer storage stability. Packing densities were measured at 0.41 g/cm^3^ for OPM and 0.47 g/cm^3^ for EPM. Higher packing density is advantageous as it reduces package volume, limits air space between particles, helps prevent oxidation, and enhances stability [33]. The angle of repose for both microcapsules was below 30°, indicating ‘good’ flowability. However, a more comprehensive analysis using the Carr index and Hausner ratio revealed distinct differences.

### 3.2. Microstructure Analysis by Scanning Electron Microscope

The morphology and surface structure of the OPM and EPM were examined using both optical microscopy and scanning electron microscopy (SEM) (Figure 1). As shown in Figure 1, SEM analysis revealed that both types of microcapsules produced by spray drying exhibited spherical shapes with relatively smooth surfaces, along with some degree of particle aggregation—common features observed in spray-dried powders. These morphological characteristics confirm the successful formation of microcapsules via the spray-drying process [34].

Optical micrographs (Figure 1a,d) provided an overview of the particle size distribution and general shape, revealing that both microcapsules were spherical but the EPM population appeared more homogeneous in size. Compared to OPM (Figure 1a), EPM (Figure 1d) displayed a more dispersed particle distribution and smaller particle sizes. This difference may be attributed to the higher solubility of EPM, which promotes better dispersion in aqueous systems and results in greater spacing between particles—an observation consistent with the solubility data presented in Table 1. Notable morphological distinctions between OPM and EPM were observed: EPM showed more surface pores and wrinkles, while OPM had a smoother and more intact outer surface. Additionally, EPM exhibited more pronounced surface depressions, and its capsule walls appeared dense with uniform thickness. In some cases, smaller microcapsules were embedded within cavities of broken shell structures. Similar morphological features have been reported in previous studies [35], which attributed these characteristics to the higher water content in EPM and the rapid evaporation of moisture during spray drying [36].

### 3.3. Particle Size Distribution and Zeta Potential of Microcapsule

The particle size distribution of OPM microcapsules, measured using a laser particle size analyzer (Figure 2A), ranged from 1.48 to 3.58 µm, with an average diameter of 2.30 µm. In contrast, EPM microcapsules exhibited a broader size range from 0.83 to 1.48 µm and an average size of 1.11 µm. Both types displayed a normal, unimodal distribution, indicating that the microcapsules were relatively uniform in size—consistent with the findings of Liu et al. [37]. The significantly smaller particle size of EPM compared to OPM suggests that EPM particles are finer and more evenly distributed, which aligns with the SEM observations shown in Figure 1.

Zeta potential reflects the magnitude of electrostatic repulsion or attraction between particles in suspension, serving as an important indicator of colloidal stability. As shown in Figure 2B, OPM exhibited a zeta potential of −36.9 mV, while EPM had a slightly lower potential of −34.6 mV. Both values fall below −30 mV, indicating that the microcapsule suspensions possess good electrostatic stability. According to Silva et al. [38], a zeta potential above ±30 mV (in absolute value) is generally considered sufficient to ensure system stability, confirming that both OPM and EPM systems are electrostatically stable.

### 3.4. FTIR Characterization of Microcapsules

FTIR analysis was conducted to further confirm the successful encapsulation of OPM and EPM. Figure 3A presents the FTIR spectra of maltodextrin (MD), soy protein isolate (SPI), camellia seed oil (OP), emulsified oil phase (EP), and the final microcapsules (OPM and EPM). Camellia seed oil exhibited several characteristic absorption peaks, with prominent bands observed in the range of 3400–3500 cm^−1^, attributed to O-H stretching vibrations. Peaks at 2935 cm^−1^ and 2852 cm^−1^ correspond to C-H stretching vibrations, while a distinct peak at 1749 cm^−1^ is associated with C=O stretching, and the band at 1466 cm^−1^ corresponds to -CH_3_ bending vibrations. The strong absorption peaks above 3000 cm^−1^ and around 1749 cm^−1^ suggest a high content of unsaturated fatty acids in camellia seed oil. The band at approximately 2852 cm^−1^ represents C-H stretching from saturated fatty acids, and the peak near 1749 cm^−1^ indicates the presence of aldehydic C=O groups, which are characteristic of oils with high unsaturation levels [39]. The emulsified phase (EP) spectrum shared the key features of OP but with a noticeable broadening of the O-H/N-H region (~3300 cm^−1^), indicating the introduction of proteins and phospholipids from the interface. Comparative analysis of the FTIR spectra of camellia seed oil, emulsified oil phase, SPI, and MD with those of the encapsulated products (OPM and EPM) revealed similar spectral patterns in key wavenumber regions. This suggests that no new chemical bonds were formed between the core and wall materials during the encapsulation process. Thus, both OPM and EPM were successfully encapsulated using SPI and MD, confirming physical embedding rather than chemical interaction.

### 3.5. Thermal Stability Analysis of Microcapsule

TGA (Thermogravimetric Analysis) and DSC (Differential Scanning Calorimetry) are complementary thermal analysis techniques. TGA was used to measure the mass change of the sample during heating, so as to determine its thermal stability and decomposition temperature. DSC was used to evaluate the thermal stability of the microcapsules by analyzing the relationship between physical or chemical changes and heat flow. When microcapsules are exposed to specific environmental conditions, temperature shifts can alter the structural state and permeability of the wall materials, gradually releasing the core substances and achieving controlled release [40].

From Figure 3B, it can be seen that the microcapsules begin to undergo phase transition at about 75 °C, which is a process of water evaporation. At the same time, the TAG curve (Figure 3C) confirms this. From room temperature to about 110 °C, it can be seen that both curves (a, b) show the first significant weight loss. The mass loss at this stage is mainly caused by the evaporation of residual water in the microcapsules. From the DSC curve, obvious exothermic peaks were observed at 173.5 °C (OPM) and 162.8 °C (EPM), which corresponded to the partial decomposition of the wall material and the transformation of the crystal structure to an amorphous form. On the TGA curve, corresponding to the temperature range of the DSC exothermic peak (about 160 °C to 410 °C), both microcapsules experienced the most severe and rapid mass loss. The weight loss at this stage is no longer due to the evaporation of water, but the decomposition of the microcapsule wall material and the release and volatilization of the core material [41]. Both of them demonstrate that the microcapsules can maintain the glassy state up to 160 °C, effectively protecting the core material, and exhibit good heat resistance. At 500 °C, the TGA curve tends to be flat. Both microcapsules have a certain amount of residue (carbon residue), and the final carbon residue rate of OPM is slightly higher than that of EPM. This part of the residue may be non-volatile inorganic components or carbonization products in the wall material or core material [42]. The difference in the amount of carbon residue may be due to the subtle difference in composition caused by the different preparation methods of the two microcapsules. The thermal response of EPM is more gentle (wider DSC peak and slower TGA weight loss step), which may mean that its wall material structure is tougher or the crosslinking degree is higher, resulting in a slower and more controllable release rate of the core material, showing the potential to be suitable as a thermally controlled release carrier.

### 3.6. Oxidation Stability of Microcapsules

The Oxidation stability index (OSI) is a widely recognized indicator for evaluating lipid stability under oxidative conditions [43]. As shown in Figure 3D, at 110 °C, the OSI of camellia seed oil (OP) increased from 3.42 h to 11.6 h after encapsulation in OPM, and further increased to 24.75 h when EP was encapsulated in EPM. These results clearly demonstrate that microencapsulation significantly enhances the oxidative stability of both OP and EP, thereby extending their shelf life. Notably, EPM exhibited superior oxidative stability compared to OPM, with the OSI increasing by 13.15 h. This enhanced stability in EPM may be attributed to its lower oil content and reduced levels of unsaturated fatty acids, which are more prone to decomposition under high temperature and oxygen-rich conditions. In contrast, the higher unsaturated fatty acid content in OPM makes it more susceptible to oxidation, leading to lower oxidative stability [44]. Furthermore, the superior oxidative stability of EPM may also be attributed to its smaller particle size (D_50_ = 1.11 μm, as shown in Figure 2A) compared to OPM (D_50_ = 2.30 μm). The higher surface area-to-volume ratio promotes a denser and more uniform wall matrix of MD and SPI, enhancing the physical barrier against oxygen and pro-oxidants, thereby effectively delaying lipid oxidation and extending the OSI [45].

### 3.7. Analysis of Lipid Composition

#### 3.7.1. Data Quality Control Analysis

To investigate the differences in lipid profiles between OPM and EPM, LC-MS/MS was employed for the qualitative and quantitative analysis of lipid species. The study focused on comparing the primary and secondary lipid classifications, fatty acid composition, and content between the two microcapsule types. The Coefficient of Variation (CV), defined as the ratio of the standard deviation to the mean of the original data, was used to assess the degree of data dispersion. A higher proportion of quality control (QC) samples with lower CV values indicates better data stability. In this study, over 85% of QC samples had CV values below 0.5, and more than 75% had CV values below 0.3, suggesting that the experimental data are highly stable and reliable. As illustrated in Figure 4A, the vertical reference lines correspond to CV values of 0.3 and 0.5, while the horizontal lines represent thresholds where 75% and 85% of the substances fall below these CV values, respectively. These findings confirm the consistency and robustness of the lipidomics data used in this analysis.

#### 3.7.2. Principal Component Analysis (PCA)

PCA was performed to preliminarily evaluate the overall metabolic differences between samples and the degree of variability within each group. The PCA results reflect the trends in lipid profile separation among sample groups, helping to identify whether significant differences exist in lipid composition between groups. The PCA score plot is shown in Figure 4B. As illustrated in Figure 4B, the combined contribution of the first and second principal components exceeds 60%, with a total contribution rate of 67.68%, indicating a meaningful and representative separation. The two sample groups are clearly separated in the plot, with no clustering between them. This suggests a high degree of dissimilarity in lipid profiles between OPM and EPM, highlighting significant differences in their lipid composition.

#### 3.7.3. Qualitative and Quantitative Analysis of Total Lipids

In this study, a total of 453 and 455 lipid species were identified in the OPM and EPM samples, respectively (Appendix A). The distinct lipid profiles of OPM and EPM, dominated by glycerolipids (GL) and glycerophospholipids (GP), are shown in Figure 5A more detailed breakdown of lipid subclasses (provided in Figure 6) confirms that the GL fraction in EPM is particularly enriched in Diacylglycerol (DG) and Monoacylglycerol (MG). According to Figure 5A, the relative abundance of lipid classes in both OPM and EPM followed the same trend: GP > GL > FA > SP > PR. Glycerolipids (GL) were predominant, accounting for 77.05% and 77.89% of the total lipid content in OPM and EPM, respectively. Further analysis of lipid composition revealed notable differences between the two types of microcapsules. Compared to OPM, the contents of triglycerides (TG), phosphatidylcholine (PC), and phosphatidylinositol (PI) in EPM were lower by 7.26%, 1.05%, and 1.5%, respectively. In contrast, the levels of diglycerides (DG), monoglycerides (MG), and free fatty acids (FFA) in EPM were higher by 6.22%, 1.84%, and 4.17%, respectively (Figure 5). These differences are consistent with the nature of the emulsion phase, which is rich in small molecules—such as DGs, MGs, and FFAs—produced through the hydrolysis of triglycerides. In contrast, TGs and phospholipids, being more lipophilic, tend to concentrate in the oil phase [46,47].

#### 3.7.4. Fatty Acid Analysis

As shown in Figure 5B, 18 and 20 types of fatty acids were detected in OPM and EPM, respectively. While the fatty acid profile (Figure 5B) indicates that EPM possesses a higher total content of unsaturated fatty acids (MUFA + PUFA = 77.37%) compared to OPM (45.63%), which would typically suggest a lower inherent oxidative stability, the OSI results (Figure 3C) demonstrate the opposite: EPM exhibited a significantly longer induction time (24.75 h) than OPM (11.6 h). The superior oxidative stability of EPM, despite its higher unsaturation degree, can be primarily attributed to its more effective physical encapsulation matrix. SEM micrographs (Figure 1) revealed that EPM possesses a denser, less porous, and more continuous wall structure—likely resulting from native emulsifiers and proteins in the emulsion phase—which forms a superior barrier against oxygen and pro-oxidants, effectively counteracting the inherent susceptibility of unsaturated lipids [45,48].

Oleic acid (C18:1) was the most abundant fatty acid in both samples. In EPM, it accounted for a notably higher proportion (57.68%) compared to OPM (27.25%). The high predominance of C18:1, a monounsaturated fatty acid, is less prone to oxidation than the polyunsaturated fatty acids (e.g., linoleic acid C18:2) that were more prevalent in OPM [49]. Thus, the combination of a less oxidation-susceptible profile of unsaturated fats collectively explains the enhanced oxidative stability of EPM. Additionally, EPM contained four unique fatty acids not present in OPM: palmitoleic acid (C16:1, 0.14%), nonenoic acid (C19:1, 0.09%), nervonic acid (C24:1, 0.08%), and docosenoic acid (C22:3, 0.01%). Palmitoleic acid (PMA), an omega-7 fatty acid, is known for its ability to regulate body weight, reduce inflammation, and improve conditions such as coronary heart disease and hypertension [50].

#### 3.7.5. Types of Triglyceride Molecules

The physiological functions of oils are not only influenced by their backbone structure but also by the type and position of fatty acids on the glycerol backbone [51]. To investigate this, the top 15 triglyceride (TG) molecules with the highest relative abundance in each sample were identified, as shown in Figure 5C. In both OPM and EPM, the most abundant TG molecules were TG (16:0_18:2_18:2) and TG (18:1_18:1_18:2), indicating that the primary TG molecular species were consistent across the two samples. In recent studies, the positional distribution of fatty acids, particularly at the sn-2 position of triglycerides, has attracted significant attention due to its relevance in digestion, absorption, and physiological functions [52]. In this study, oleic acid (18:1) and linoleic acid (18:2) were the dominant fatty acids at the sn-2 position, with oleic acid being the primary component-consistent with its high abundance in camellia seed oil. This composition enhances the oil’s digestibility and nutritional value, thereby contributing to the health-promoting effects of camellia seed oil [53]. Figure 5C also shows that OPM contained three additional TG species not detected in EPM: TG (18:1_18:1_20:1), TG (18:1_18:2_18:2), and TG (18:1_18:1_18:3), with relative abundances of 6.89%, 6.42%, and 6.29%, respectively. These TG molecules predominantly feature oleic acid (18:1) in their fatty acid chains. In OPM, oleic acid primarily exists in the form of triglycerides, diacylglycerols, or monoacylglycerols, with relatively little present in the free fatty acid form. This phenomenon may be attributed to the polymerization of free fatty acids into glycerol-based lipids during the microencapsulation process, which aligns with the lower free oleic acid content observed in the fatty acid composition analysis in Section 3.7.3. Similar findings have been reported by V. et al. [54], supporting this interpretation.

## 4. Conclusions

In this study, EPM was prepared by spray drying a mixture of EP, SPI, and MD, and then compared with OPM. Both OP and EP were successfully encapsulated, achieving encapsulation efficiencies (EE) of 86.53% and 83.94%, respectively. Notably, EPM outperformed OPM in solubility, bulk density, and particle size. SEM images revealed that the microcapsules were regularly spherical with smooth surfaces, although some particle aggregation was observed. FTIR analysis confirmed the successful encapsulation of both the oil phase and the emulsion phase, while differential scanning calorimetry demonstrated the high thermal stability of the microcapsules. Particularly under continuous heating at 110 °C, the oxidation stability of EPM was significantly higher than that of OPM, reaching 2.13 times that of OPM. Qualitative and quantitative lipid analysis identified 447 lipids across 5 primary categories and 28 subclasses in both OPM and EPM. Significant differences were observed in the content and types of free fatty acids (FFA) and triglycerides (TG) between the two groups. Although this study successfully developed emulsion-phase microcapsules (EPM), their stability, solubility, and other functions are superior to those of traditional oil-phase microcapsules (OPM), highlighting the potential of EPM as a sustainable functional food component. However, this study also has certain limitations. Therefore, future research will address limitations such as in vivo bioavailability, release kinetics, and scalability to facilitate practical application.

## Figures and Tables

**Figure 1 foods-14-03314-f001:**
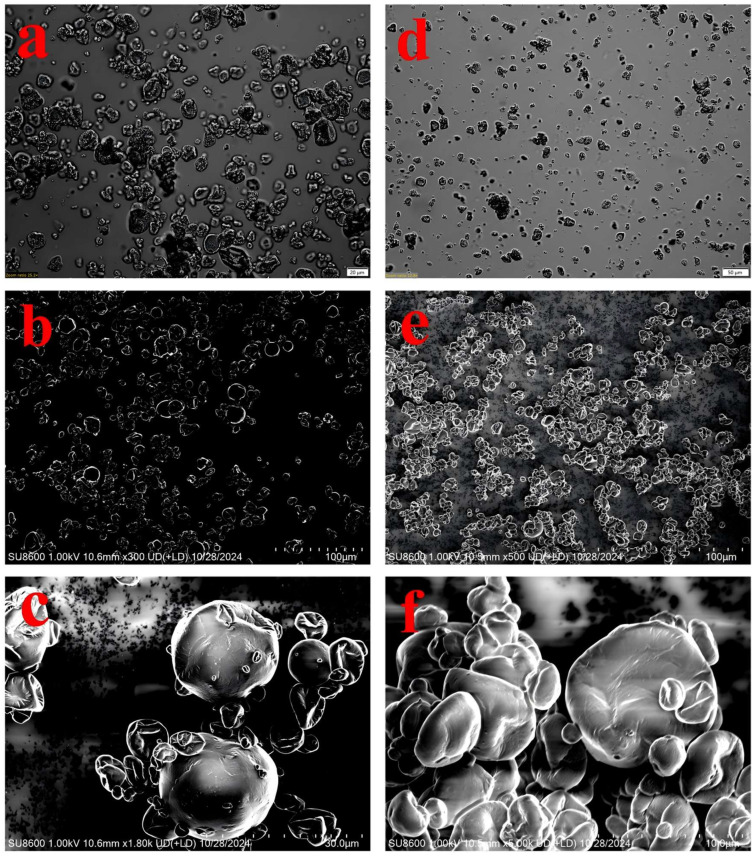
Microscopic images of OPM × 20 μm (**a**), EPM × 50 μm (**d**) and SEM images of OPM × 100 μm (**b**), OPM × 30 μm (**c**), EPM × 100 μm (**e**), EPM × 10 μm (**f**).

**Figure 2 foods-14-03314-f002:**
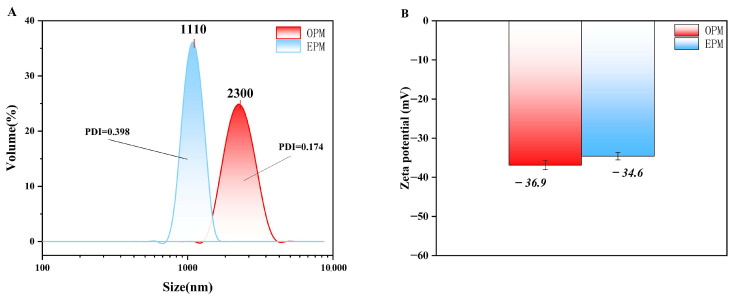
Particle size distribution (**A**) and Zeta potential (**B**) of microcapsules.

**Figure 3 foods-14-03314-f003:**
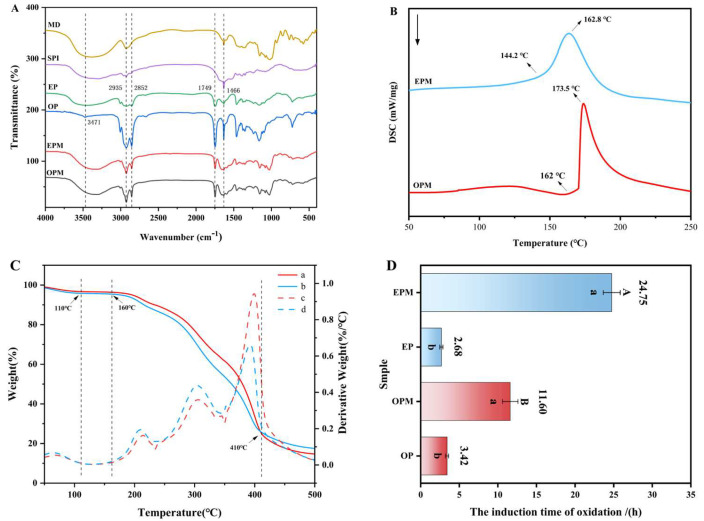
FTIR spectra of MD, SPI, OP, EP, OPM and EPM (**A**); Differential scanning calorimetry curves of OPM and EPM (**B**); The thermogravimetric and differential thermal analysis curves of OPM (a, c) and EPM (b, d) (**C**); Oxidation stability of OP, EP, OPM, and EPM (**D**). In (**D**), a and b were *t*-test analyses of data before and after microencapsulation; (**A**,**B**) were *t*-test analyses between OPM and EPM (*p* < 0.05).

**Figure 4 foods-14-03314-f004:**
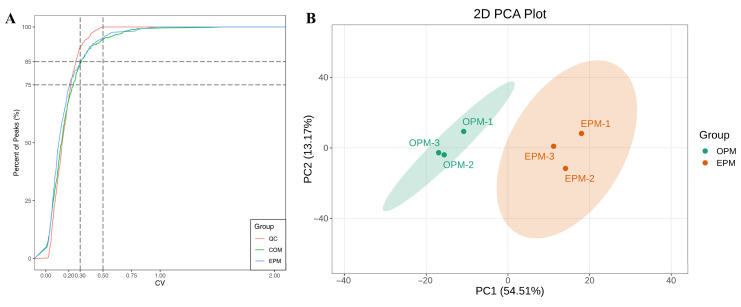
CV distribution in each group of samples (**A**); PCA score chart of each sample quality spectrum data (**B**).

**Figure 5 foods-14-03314-f005:**
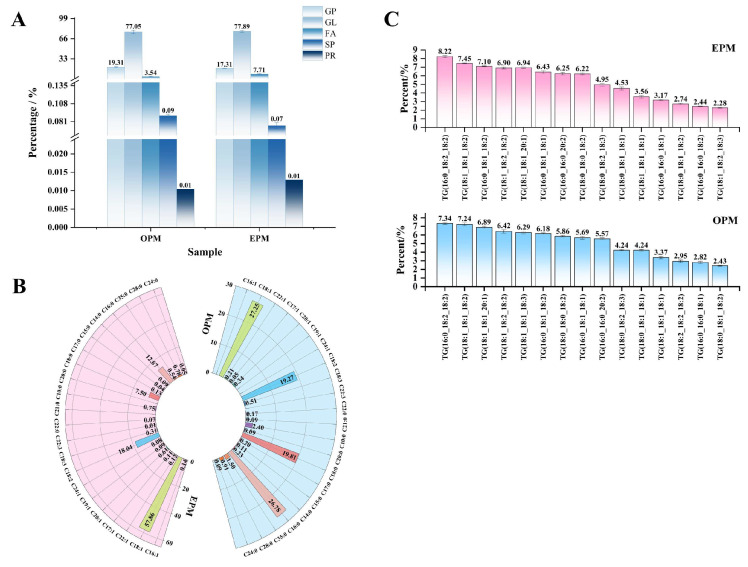
Histogram of percentage lipid subclasses (**A**); Fatty acid composition and content ring diagram (**B**); Triglyceride type histogram (**C**).

**Figure 6 foods-14-03314-f006:**
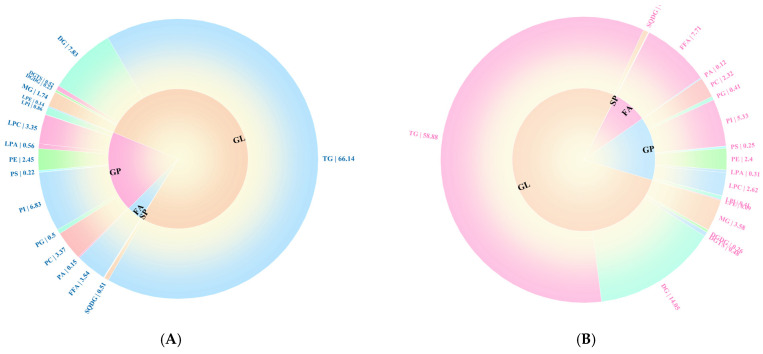
Lipid composition of OPM (**A**) and EPM (**B**) secondary classification percentage loop (<0.1 not shown).

**Table 1 foods-14-03314-t001:** The physical properties of microcapsules.

	OPM	EPM
Visual and Textural Observations		
color	milky white	light gray
Taste and smell	free from extraneous odor	free from extraneous odor
texture	powder-like, loose, no caking	powder-like, loose, no caking
Color parameters		
*L**	95.90 ± 0.8 ^b^	96.22 ± 0.5 ^a^
*a**	−0.3 ± 0.01 ^b^	0.17 ± 0.01 ^a^
*b**	2.67 ± 0.08 ^b^	2.82 ± 0.03 ^a^
Physical properties		
EE (%)	86.53 ± 0.14 ^a^	83.94 ± 0.1 ^b^
Moisture content (%)	1.29 ± 0.03 ^b^	1.88 ± 0.02 ^a^
Dispersibility (%)	57.68 ± 0.05 ^b^	90.50 ± 0.09 ^a^
Carr index (%)	18.2 ± 1.5 ^a^	12.5 ± 1.1 ^b^
Hausner ratio	1.22 ± 0.02 ^a^	1.14 ± 0.02 ^b^
Angle of repose (°)	12.73 ± 0.01 ^a^	10.56 ± 0.01 ^b^
Bulk density (g/cm^3^)	0.41 ± 0.01 ^b^	0.47 ± 0.01 ^a^

Note: OPM means the oil-phase microcapsules; EPM means the emulsion-phase microcapsules. Value reported is mean ± standard deviation (*n* = 3). Different letters indicate significant differences (*p* < 0.05).

## Data Availability

The original contributions presented in this study are included in the article/Appendix A. Further inquiries can be directed to the corresponding author.

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
