# Peer review of "Microencapsulation of Camellia oleifera Seed Oil Emulsion By-Products: Structural Characterization and Lipidomics Analysis"

_foods, 2025, doi:10.3390/foods14193314_

Round 1
Reviewer 1 Report
Comments and Suggestions for Authors
1. The title could be more specific; in its current format, it is too general.
2. In the abstract, it is recommended not to overuse abbreviations and to improve the introductory text in lines 11-12 and the final lines 24-26 to conclude its application in the food industry better.
3. It is recommended to use MDPI citations and numerical references, using the ENDNOTE PLUG IN.
4. Ensure that all abbreviations are explained the first time they appear in the paragraph. It is recommended to include a section on abbreviations before the references.
5. In the introduction, include the knowledge gap that the study aims to fill, and explicitly state the hypotheses. It is highly recommended that the novelty and purposes of the study be presented at the end of the introduction in bullet points.
6. Ensure that all inputs and reagents are included in the materials and methods. Verify the same with all equipment used in the research, including (model, brand, city, and country) if applicable.
7. Does section 2.3.1 not require an equation for the calculation?
8. In lines 173 and 174, ensure that the meters in the equation have the same format as the notes.
9. Provide as many details as possible about the sensory evaluation in section 2.4.1. Section 2.4.2 lacks a citation. Both sections should have more methodological details.
10. In line 195, standardize the period for the comma in the magnification (according to the international system).
11. Why was TGA not measured to complement DSC?
12. Why do sections 2.8.1 and 2.8.3 not have sources?
13. Section 2.9 should explicitly state which experimental design was used and which statistical tests were used for that design, as well as include all the statistical assumptions that were evaluated. In addition, include all programs used, for example, for graphical representation (e.g., the one used to make Figure 2 and the others). Also include their licenses and additional details in parentheses.
14. Improve the presentation of Table 1, which is not visually appealing and is somewhat confusing in its current format. The sensory results should be processed separately (is there no non-parametric statistic for this point?).
15. In the encapsulation efficiency (EE) results, there is an inconsistency in values (83.00% vs. 83.94%) that should be unified.
16. Regarding particle size, it is stated that EPM has a smaller size, but the data show the opposite; correct for consistency.
17. The letters in Figure 1 should be in parentheses (the micrographs lack a scale bar and a clear description of morphology).
18. Only the angle of repose is reported; include other indices (Carr, Hausner) to improve interpretation.
19. There are wave numbers that have not been interpreted in Figure 3a that would be interesting to include and discuss (in FTIR, review the assignment of bands and functional groups, for example, at 1749, are they esters or aldehydes?).
20. Check the oxidative stability (OSI), review the different values and temperatures between the summary, results, and conclusions (unify).
21. In Figure 3b (DSC), what is the glass transition temperature? On the other hand, the behavior of the red line of the OPM seems strange.
22. In Figures 5 and 6, some letters are illegible, so it is recommended to improve the graphic representation, as well as to use a single color palette throughout the document.
23. Discrepancy in the number of species (477 vs. 453/455); clarify counting criteria in the lipodomics section.
24. Why were no in vitro release tests performed, and why were the microcapsules obtained not included in food matrices?
25. It is recommended to use citations from the last 5 years. In addition, the discussions should include the physical, chemical, and microbiological mechanisms involved. Do not just make comparative discussions. Also discuss whether its application in the food industry is possible, considering costs and other important aspects.
26. The conclusions should be improved, including the limitations of the study and possible new lines of research. The potential for application in the food industry should also be highlighted.
27. It is recommended to reduce the iThenticate similarity index (26%), especially in the materials and methods section.
Reviewer 2 Report
Comments and Suggestions for Authors
The manuscript discusses the preparation of microcapsules from camellia oleifera seed oil emulsions using the spray drying technique. The authors studied and compared two microcapsules obtained from emulsion systems prepared with the by-products (EPM) and the oil (OPM). The topic of the manuscript suits the journal scope and should be of interest to its readers. However, the manuscript should be extensively improved in the Methodology section as necessary information is currently missing and not provided to the reader. In the Results section, additional discussions should be provided in some sections and the authors should pay a specific attention to accurate presentation of discussions based on the provided figures. In some sections, the presented figures remain undiscussed and the inclusion of discussions is required. A number of references are not accurately presented and should be corrected.
I have the following comments to the authors:
- In lines 11-15, the sentence is too long and difficult to read. break this section into shorter sentences.
- Provide a reference for the statement provided in lines 34-35.
- In section 2.3.2:
-
- line 158, what does 4g (m) present? The m in parenthesis is not clear.
- In equation 2, “oli” should be corrected to “oil”
- In lines 173-174, what do authors mean by "quality"? Do they intend to refer to the mass? Please correct.
- In section 2.4.3, provide the specifications and brand name of the microscope used. Provide additional details in this section and cite another adequate reference.
- In section 2.4.4, what was the working distance for SEM analysis?
- In section 2.4.5, clarify what does “M” refer to in the equation. Clarify why the gram of powder in the petri dish was multiplied by 3.
- In section 2.5, what was the resolution for the FTIR analysis?
- In section 2.7, line 242, provide the specifications and brand name of the used device (oxidation stabilizer).
- In section 2.8.2, lines 257-258, provide the specifications and brand name of the UPLC device coupled with tandem mass spectrometer.
- In Table 1, L should be corrected to L*.
- In section 3.3, lines 350-355, a reverse trend of particle size is shown in the text not consistent with Figure 2A. Please check and present accurately.
- In Figure 2, the colours of OPM and EPM in the “figures 2a and b” are not consistent. Present OPM as blue and EPM as red in both or the other way around.
- In section 3.6, lines 418-422, What about its particle size? Please add additional discussion to the main text on the impact of EPM particle size on its stability.
- In line 430, provide the complete term for QC.
- Figure 6 is unexplained in the text. Provide discussions on the reported results presented in Figure 6.
- In section 3.7.3, based on the information presented in Figure 5B, OPM has higher saturated fatty acids (C16:0) and (C18:0), and unsaturated fatty acids (C18:2) compared to EPM whereas the content of MUFAs (C18:1) is higher in EPM. Add additional discussion to the main text on how the composition of fatty acids can impact the oxidative stability of EPM compared to OPM and provide explanations based on the results presented in Figure 3C. Figure 3C shows higher oxidation induction time for EPM although it has higher MUFA and PUFA content as the authors denoted in lines 465-468.
- Figures should be cited consecutively in the main text. Section 3.7.5 should be presented as section 3.7.2 after section 3.7.1. Please rearrange.
- A number of references should be corrected and presented accurately in the main text and the reference list. The last name of the authors are not correctly captured.
- Line 568, E.X.
- Line 606, M.L.V.
- Line 633, S.S.
- Line 636, The author name “Sining X.” should be corrected to Xu, S.. Present the author names with their last names.
- Line 652, V.R.C.
Check similarly throughout the text and correct if required.
Comments on the Quality of English LanguageThe sentence structure should be improved (please see my comments).
Reviewer 3 Report
Comments and Suggestions for Authors
The manuscript contains an interesting study on a timely topic, both with respect to using enzymatic rather than solvent extraction and with respect to valorization of rest raw materials. I do however have some concerns and questions:
1) The study is based on two different microcapsules - one based on the emulsion phase and one based on the oil phase. Is this sufficient to reach the conclusions presented in this manuscript?
2) In my opinion, the use of the term "sensory evaluation", including "taste and smell" in table 1 is overstating the presented results. If a standardized sensory evaluation was performed, more details should be given, including the size of the test panel, etc, as opposed to a qualitative evaluation of the smell (without a robust reference material) and appearance.
3) The description of enzymatic extraction should at the very least provide details on what enzyme was used.
4) Please provide more details about the principal component analysis - both in terms of method (e.g. how was the data set standardized, what were the dimensions of the dataframe, how many principal components were selected initially etc) and in terms of the results obtained (e.g. loadings and residual plots). Absent further information, it is difficult to assess whether there is sufficient data (in terms of rows/samples) to warrant the use of PCA, despite the promising score plot shown in Figure 4b.
5) Given the measured size of the capsules, laser diffraction would be a much better choice than DLS for analysis of mean size. Both from the size distribution curves and from the SEM images, the microcapsules are at the very limit of what can be accurately measured using thesetup listed.
6) If microcapsules are soluble, it means that they disintegrate. Please change "solubility of microcapsules" to "dispersibility of microcapsules" in the text
Round 2
Reviewer 1 Report
Comments and Suggestions for Authors
- Although the authors claim to have determined the glass transition temperature (Tg) using DSC, in Section 3.5 and Figure 3B, a clear Tg is not presented. What is observed are changes associated with water evaporation (~75 °C) and exothermic peaks at 173.5 °C and 162.8 °C, which correspond more closely to melting or partial decomposition of the matrix rather than a glass transition. Tg is typically characterized by a subtle baseline shift, rather than well-defined peaks, as reported by the authors. Their interpretation is therefore not well supported, and the absence of complementary TGA analysis limits the accurate characterization of thermal stability. It is recommended to clarify this point, properly identify the Tg, and, if possible, complement the results with TGA to differentiate between moisture loss, glass transition, and thermal decomposition.
- The authors removed the sensory evaluation because a formal panel did not conduct it, but rather involved basic physical descriptions. However, instead of suppressing this information, they should have clarified it as a visual or physical characterization. It is recommended to reincorporate these data under an appropriate title, clearly distinguishing them from a validated sensory analysis. While avoiding a misleading term such as “sensory evaluation” is understandable, this information should not have been entirely omitted but rather explicitly clarified in the manuscript.
- The manuscript lacks a clearly written final section on limitations. The authors only justify the absence of TGA but do not discuss other important constraints, such as experimental scale, industrial applicability, or the absence of release and long-term stability studies.
- The statistical analysis is basic and does not adequately address the multiple variables assessed. More robust statistical methods, such as multifactorial ANOVA, PCA, or cluster analysis, should be applied to strengthen the validity of the results and conclusions.
Reviewer 2 Report
Comments and Suggestions for Authors
My previous comments to the authors are mostly addressed. Still, the authors should pay a specific attention to the correction of the following existing mistakes:
- In lines 183-185, do authors intend to mention M1 and M2 presented in equation 4? Currently W1 and W2 are used mistakenly. Please correct.
- In line 207, do authors intend to mention equation 5 in the text? Please check and correct.
- Equation numbering should be corrected. In line 228, equation 5 should be corrected to 8. In line 298, equation 5 to 9. Check if they are also accurately cited in the main text.
- Still, mistakes exist in the presentation of references. Although I requested their correction in the previous round of peer-review. The author names are incorrectly captured in references [8], [32], [49], and [51]. They are abbreviated or presented with the first name by mistake (in reference [32]). They should be checked and corrected. Please check also all other references and implement any necessary corrections.
Reviewer 3 Report
Comments and Suggestions for Authors
The authors have addressed all comments and concerns
Author Response
Comment: The authors have addressed all comments and concerns.
Response: We are deeply grateful to the reviewer for this positive assessment and for their time and expertise throughout the review process. We are pleased that our revisions have fully addressed the previous comments to the reviewer's satisfaction.